# Impact of Ethanolic Thai Indigenous Leaf Extracts on Melanosis Prevention and Shelf-Life Extension of Refrigerated Pacific White Shrimp

**DOI:** 10.3390/foods12193649

**Published:** 2023-10-02

**Authors:** Abubakar Saleh Ahmad, Thanasak Sae-leaw, Bin Zhang, Prabjeet Singh, Jun Tae Kim, Soottawat Benjakul

**Affiliations:** 1International Center of Excellence in Seafood Science and Innovation, Faculty of Agro-Industry, Prince of Songkla University, Hat Yai 90110, Songkhla, Thailand; ahmadabubakarsale@gmail.com (A.S.A.); thanasak.s@psu.ac.th (T.S.-l.); 2Key Laboratory of Health Risk Factors for Seafood of Zhejiang Province, College of Food Science and Pharmacy, Zhejiang Ocean University, Zhoushan 316022, China; zhangbin@zjou.edu.cn; 3College of Fisheries, Guru Angad Dev Veterinary and Animal Sciences University, Ludhiana 141004, Punjab, India; prabjeetsingh@gadvasu.in; 4Department of Food and Nutrition, Kyung Hee University, Seoul 02447, Republic of Korea; jtkim92@khu.ac.kr

**Keywords:** soursop, Pacific white shrimp, anti-melanosis, polyphenoloxidase, quality, shelf-life

## Abstract

Shrimp has been known for its delicacy, but it undergoes rapid deterioration induced by biochemical and microbiological reactions. Melanosis is a major cause of discoloration associated with consumer rejection. All ethanolic extracts from different leaves including soursop, noni, and Jik leaves were dechlorophyllized via the “Green” sedimentation method before being used. The inhibitory activity against polyphenoloxidase (PPO) from Pacific white shrimp (*Litopeneous vannamei*) and the copper-chelating properties of varying extracts were compared. Soursop leaf extract (SLE) showed higher PPO inhibitory activity and copper-chelating ability than others (*p* < 0.05). Based on LC-MS, aempferol-3-O-rutinoside was identified as the most abundant compound, followed by catechin and neocholorigenic acid. The efficacy of SLE at different levels (0.25–1%) for inhibiting melanosis and preserving the quality of Pacific white shrimp was evaluated during refrigerated storage at 4 °C for 12 days in comparison with that of a 1.25% sodium metabisulfite (SMS)-treated sample. SLE at a level of 1% effectively retarded melanosis and bacterial growth, in which the total viable count did not exceed the microbial limit within 12 days. In addition, 1% SLE treatment impeded autolysis, reduced protein degradation and decomposition, and minimized lipid oxidation, as witnessed by the lower increases in pH, TVB-N, and TBARS values. Sensory evaluation indicated higher likeness scores and overall acceptability for SLE-1% and SMS-1.25% shrimps than those of the control and other samples. Therefore, SLE could be used as a natural alternative that effectively lowered the melanosis and quality loss of shrimp during refrigerated storage.

## 1. Introduction

Pacific white shrimp (*Litopenaeus vannamei*) is a highly valued farmed species due to its delicacy and high protein content. Its commercial value has increased up to 20.6 billion USD [1,2] with an estimated annual production of 5.8 million tons [3]. However, it is highly perishable and prone to spoilage. In addition, melanosis or black spots developed on the normally white or translucent shrimp body surface during post-harvest storage is an inevitable problem for shrimp farmers and traders since this phenomenon brings about low acceptability by customers [4]. Melanosis is induced by polyphenoloxidase (PPO), which causes the oxidation of phenolic compounds into quinones [5] with subsequent non-enzymatic polymerization into a high-molecular-weight dark pigment called “melanin”, which is generally developed under the carapace of shrimp cephalothorax [6,7]. Melanosis varies with species, catching and handling procedures, and the physiological conditions of the crustaceans [8]. Lipid oxidation also leads to quality loss in shrimp because of its high level of polyunsaturated fatty acids (PUFA). Autoxidation and enzymatic oxidation induced by lipoxygenase and peroxidase directly contribute to the off-flavor. Although melanosis does not cause harm to consumers [9], it negatively affects the sensory attributes and overall quality of crustaceans.

The prevention of melanosis and other quality deteriorations in shrimp entails the use of several preservation techniques including pre-cooking, refrigeration, modified-atmosphere packaging, and the use of chemicals like sodium metabisulfite (SMS) and 4-hexylresorcinol (4-HR) [4,10,11]. However, in the seafood industry, natural additives have drawn attention to prevent melanosis in crustaceans owing to consumer awareness regarding the risks of chemical preservatives. Sulfites can cause a range of allergic reactions, such as hives, itching, flushing, breathing difficulties, and rarely anaphylaxis [12]. The use of 4-HR has some limitations because it is still expensive to apply on a commercial scale. To provide an alternative solution, natural additives such as plant extracts can be the better choice. Several plant extracts have shown promising anti-melanosis, antioxidant, and antimicrobial effects on a variety of crustaceans. Chamuang leaf extract [13], cashew leaf extract [11], guava leaf extract [14], lead seed extract [7], and pomegranate peel extract [15] were applied successfully to retard the microbiological and chemical spoilage as well as to inhibit melanosis in shrimp during cold storage. Polyphenols such as ferulic acid and cinnamic acid as well as flavonoids have shown appreciable melanosis inhibition and antioxidant properties [16,17].

Thailand has numerous plants whose leaves can be used as food or folk medicine [18]. Soursop (*Annona muricata* L.), a tropical fruit plant, is widely cultivated for its delicious fruit, which has a unique sweet and sour taste. The fruit also has a creamy texture, while the leaves have been traditionally used to treat illnesses such as fever, headache, diarrhea, respiratory infections, ulcers, hypertension, cancer, and diabetes [19,20]. The major bioactive compounds in soursop include alkaloids, acetogenins, glycosides, polyphenols, flavonoids, vitamins, and tannins, etc. [19,21]. The potentialities of the leaf, root, bark, stem, and seed extracts from soursop have been demonstrated as good antimicrobial and antioxidative agents. Correa-Gordillo et al. [22] documented the antioxidant activity of fresh and frozen pulp, juice, and fresh or dried soursop leaves. In another study, soursop extract showed appreciable bactericidal and bacteriostatic activities against *Staphylococcus aureus*, *Bacillus cereus*, *Enterococcus faecalis*, *Enterobacter aerogenes*, *Enterobacter cloacae*, *Pseudomonas aeruginosa*, *Escherichia coli*, *Salmonella typhimurium*, *Salmonella choleraesuis*, and *Shigella dysentery* [23]. Noni (*Morinda citrifolia*) is a tropical plant that belongs to the Rubiaceae family and is widely distributed in the Pacific Islands, Southeast Asia, and Australia. Noni leaf, root, and fruit juice have been consumed as traditional medicine for centuries due to their significant health benefits [24]. The leaf possesses various beneficial properties including anti-inflammatory, antioxidant, and antimicrobial activities [25]. In addition, noni leaf also exhibits anticancer and immunomodulatory properties [26]. These bioactivities are linked to the presence of phenolic compounds in the plant. Jik (*Barringtonia acutangula*), an evergreen plant belonging to the Lecythidaceae family, is commonly known as a freshwater mangrove. It is available across southeast Asia, east Africa, and northern Australia. In Thailand, “Jik” is the most abundant species among the country’s 11 commonly found species of Barringtonia [27]. The plant has been used for centuries in traditional medicine to treat a variety of ailments, including fever, inflammation, constipation, and diarrhea [28]. Jik stem bark, containing secondary metabolites like tannins, flavonoids, terpenes, sterols, and phenolic compounds, has anti-diabetic, antimicrobial, and antioxidant activity [29]. Recently, the leaves of three indigenous plants in southern Thailand, namely noni (*Morinda citrifolia*), Jik (*Barringtonia acutangula*), and soursop (*Annona muricata*), were extracted using ethanol at various concentrations and dechlorophyllized using a sedimentation method. Those extracts showed varying antioxidant and antimicrobial activities [30]. However, the information on the PPO inhibitory activity (PPO-IA) and copper-chelating activity (CCA) of those extracts was scarce. This study aimed to investigate PPO-IA and CCA of various dechlorophyllized ethanolic leaf extracts and to elucidate the preservatory role of the selected extracts showing high PPO-IA in the shelf-life extension of refrigerated Pacific white shrimp.

## 2. Materials and Methods

### 2.1. Chemicals

All chemicals used were of analytical grade. Thiobarbituric acid (TBA), trichloroacetic acid (TCA), 1,1,3,3-tetramethoxypropane, Brij-L23, tetramethylmurexide, and L-β-(3,4 dihydroxyphenyl) alanine (L-DOPA) were procured from Sigma Aldrich (St. Louis, MO, USA). Potassium carbonate and copper (II) sulfate were purchased from Thermo Fisher Scientific (Auckland, New Zealand). The microbial media (eosin methylene blue agar (EMB), *Pseudomonas* isolation agar (PIA), plate count agar (PCA), triple sugar iron agar, and Mueller–Hinton broth (MHB) were acquired from Himedia (Mumbai, India).

### 2.2. Collection of Leaves and Preparation of Leaf Extracts

Three Thai indigenous leaves including soursop (*Annona muricata*), noni (*Morinda citrifolia*), and Jik (*Barringtonia acutangula*) were obtained from a plantation in Hat Yai, southern Thailand. The leaves were sorted, washed, and dried at 60 °C in an oven to obtain a constant weight. The dried leaves were powdered using a high-speed grinder (Panasonic, model MX-898N, Berkshire, UK) and subsequently sieved using a stainless-steel sieve (80 mesh). The powders were named NLP, JLP, and SLP for noni, Jik, and soursop leaf powder, respectively.

To prepare the extracts, the leaf powder (50 g each) was added with 500 mL of 80% ethanol and stirred using a magnetic stirrer at a medium speed for 60 min. The centrifugation of the mixture (3000× *g* for 30 min) was performed using a refrigerated centrifuge (Hitachi, model CR22 N, Tokyo, Japan). Whatman filter paper No. 1 was then employed to filter the supernatant. The ethanol was removed from the filtrate using a rotary evaporator (Eyela, Rikakikai Co. Ltd., Tokyo, Japan) at 40 °C. The resulting extracts were dechlorophyllized using the sedimentation method [31]. To the extract, distilled water was added at a 1:2 ratio (*v*/*v*), stirred, and left at 4 °C for 24 h. The supernatant layer (without chlorophyll) was centrifuged (10,000× *g*, 4 °C for 30 min), filtered, and freeze-dried. The dechlorophyllized extract powders were named NLE, JLE, and SLE for dechlorophyllized ethanolic noni, Jik, and soursop leaf extracts, respectively.

### 2.3. PPO Inhibition and CCA of Leaf Extracts

#### 2.3.1. Extraction of PPO from Cephalothoraxes of Pacific White Shrimp

Pacific white shrimp cephalothorax powder was first prepared, and PPO was extracted [11]. The powder was mixed with three volumes of extracting buffer (50 mM sodium phosphate buffer, pH 7.2 containing 1.0 M NaCl and 0.2% Brij-L23). The mixtures were continuously stirred for 30 min and then centrifuged (8000× *g*, 30 min) using a refrigerated centrifuge. Ammonium sulfate fractionation using 40% saturation was carried out and the pellet was collected via centrifugation (12,500× *g*, 30 min). Thereafter, 10 mL of 50 mM sodium phosphate, pH 7.2, was added into a pellet, followed by dialysis using 50 volumes of the same buffer for 18 h. Buffer replacement was performed at 6 h intervals. Finally, the centrifugation (3000× *g*, 30 min) of dialysate was done and the obtained supernatant was named “PPO extract”.

#### 2.3.2. Measurement of PPO Activity

PPO activity was determined using DOPA as a substrate [32]. The assay was performed by adding 20 μL of PPO extract to 120 μL of 15 mM L-DOPA in distilled water, 80 μL of 50 mM phosphate buffer (pH 6.0), and 20 μL of distilled water. After incubation (45 °C, 3 min), the absorbance at 475 nm was read using a microplate reader (FLUOstar Omega, model: 415-101, BMG LABTECH, Ortenberg, Germany). One unit of PPO activity meant the enzyme yielded an increase in the absorbance at 475 nm by 0.001 per min. Enzyme and substrate blanks were prepared by omitting the substrate and enzyme, respectively, from the reaction mixture. Distilled water was used instead.

#### 2.3.3. Measurement of PPO-IA

PPO extract (20 μL) was mixed with the leaf extracts (20 μL) in a 96-well microplate to obtain the final concentrations of 0.125, 0.25, 0.5, and 1%. After incubation at room temperature (26 ± 1 °C) for 30 min, 80 μL of assay buffer (50 mM sodium phosphate, pH 6.0) was added. Subsequently, 120 μL of pre-incubated 15 mM L-DOPA solution (45 °C) was added and incubated at 45 °C for 3 min and the absorbance at 475 nm was recorded [11]. The control was also prepared [11]. PPO-IA was computed as follows:$$Inhibition\left(\%\right)=\left[\frac{A-B}{A}\right]\times 100$$
where A is the PPO activity of the control and B is the PPO activity in the presence of a leaf extract.

#### 2.3.4. Copper-Chelating Activity (CCA)

The CCA of the leaf extracts was examined [25]. To 1 mL of leaf extract solution, 1 mL of solution (1 mM Cu_2_SO_4_ in 10 mM hexamine–HCl buffer containing 10 mM KCl, pH 5.0) was added and left for 10 min. Thereafter, 100 μL of 1 mM tetramurexide in the same buffer was added. The final concentrations of the leaf extracts were 0.125, 0.25, 0.50, and 1.0%. The absorbance ratio (A_460_/A_530_ nm) was calculated and converted to the corresponding free copper (II) ion concentration with the aid of the standard curve of free copper (II) ion concentration (50–400 mM) versus absorbance ratio. The difference between the total copper (II) ion and free copper (II) ion concentrations indicated the copper (II) ions chelated by the extracts. The CCA was computed using the following equation:$$CCA\left(\%\right)=[\mathrm{Concentration}\;\mathrm{of}\;\mathrm{chelated}\;{Cu}^{2+}/\;\mathrm{Concentration}\;\mathrm{of}\;\mathrm{total}\;{Cu}^{2+}]\times 100$$

SLE, yielding the highest anti-melanosis activity, as indicated by the highest PPO-IA and CCA, was chosen for further study.

### 2.4. Analysis of Phenolic Compounds in Soursop Leaf Extract (SLE)

The identification of compounds in SLE was carried out using liquid chromatography/mass spectrometry (LC-MS/QTOF X500R) using negatively ionized electrospray (ESI) mode as detailed by Chotphruethipong et al. [33]. The phenolic compounds present were classified and expressed as abundance.

### 2.5. Study on the Effects of Soursop Leaf Extract (SLE) on the Quality of Pacific White Shrimp during Refrigerated Storage

#### 2.5.1. Preparation of Pacific White Shrimp Treated with SLE at Various Concentrations

Freshly caught Pacific white shrimps (55–60 shrimps/kg) purchased from a local market in Hat Yai, Thailand, and kept in a polystyrene box with flaked ice (shrimp/ice ratio of 1:2 (*w*/*w*)) were transported to the laboratory within 30 min. After washing using tap water and draining on the screen for 3 min, whole shrimps were immersed into the SLE solutions at varying levels (0.05, 0.1, 0.25, 0.5, and 1%) using a solution/shrimp ratio of 2:1 (*v*/*w*) and then kept at 4 °C for 30 min. Another group of whole shrimps was treated with a 1.25% SMS solution and kept under the same conditions. The control group was prepared in the same way by immersing the shrimp in distilled water and kept at the same temperature and time. After the treatments, the samples were drained using a stainless-steel screen, transferred to polystyrene trays, wrapped with shrinkable plastic film, and kept at a refrigerated temperature (5 ± 1 °C). The samples were randomly taken for analysis at 3-day intervals for 12 days.

#### 2.5.2. Melanosis Assessment

Melanosis was assessed through visual inspection by 10 trained panelists using a 10-point scoring test [32]. Panelists were instructed to give a melanosis score (0–10), where 0 = absent and 10 = extreme (100% of shrimp surface affected).

#### 2.5.3. Microbiological Quality

A microbiological assay was performed on all samples following the method of Shiekh et al. [13]. The spread plate method was used to measure total viable count (TVC), psychrotrophic bacterial count (PBC), *Pseudomonas* count, *Enterobacteriaceae* count, and H_2_S-producing bacteria count. Shrimp meat (10 g) was mixed with 90 mL of 0.85% sterile saline solution and homogenized using a Stomacher (400 Circulator, Seward Ltd., West Sussex, UK) at 220 rpm for 2 min. Serial 10-fold dilution was carried out for homogenate using 0.85% sterile saline solution. PCA was used to determine TVC and PBC (37 °C for 2 days and 4 °C for 10 days, respectively). *Pseudomonas* and H_2_S-producing bacteria count were enumerated by the presence of black colonies on triple sugar iron agar and white colonies on *Pseudomonas* isolation agar, respectively, after 3 days of incubation at 25 °C. *Enterobacteriaceae* was cultured on eosin methylene blue (EMB) agar, as indicated by the dark colonies with a green metallic luster after incubation at 37 °C for 1 day.

#### 2.5.4. Total Volatile Base (TVB) Content

The TVB-N content of the stored shrimp meat was analyzed as per the method employed by Sae-leaw and Benjakul [11]. The TVB-N content was evaluated and reported as mg N/100 g of shrimp meat.

#### 2.5.5. pH

A shrimp meat sample (2 g) was homogenized (8000 rpm, 1 min) with deionized water (20 mL). The pH of the homogenate was examined using a pH meter (Eutech 700, Thermos Fisher Scientific Inc., Waltham, MA, USA) [34].

#### 2.5.6. Thiobarbituric Acid Reactive Substances (TBARS)

The TBARS of the stored shrimp sample were determined as described by Shiekh and Benjakul [35] using a TBA reagent. The absorbance of the reaction mixture was read at 532 nm using a spectrophotometer (model UV-1800, Shimadzu, Kyoto, Japan). TBARS values were calculated from the standard curve of malonaldehyde (0–2 ppm) and reported as mg malonaldehyde/kg shrimp meat.

#### 2.5.7. Sensory Evaluation

Whole Pacific white shrimps at days 0 and 12 of refrigerated storage (with the microbial load under the limit) were placed in a stainless-steel cooking pot containing boiling water (shrimp/water ratio, 1:3 *w*/*v*). The shrimps were pre-cooked until the core temperature reached 70 °C and held for 30 s. The sample was rapidly cooled to room temperature using iced water and drained using a stainless-steel screen before serving. Sensory evaluation was conducted by 50 untrained panelists comprising staff and students from the Food Science and Technology program, Prince of Songkla University. The panelists were regular consumers of shrimp and had no allergies to shrimp. Appearance, color, texture, taste, flavor, odor, and overall likeness of the sample were evaluated using a 9-point hedonic scale [36]. The samples were coded in three-digit random numbers and presented on paper trays to the panelists. Sensory analysis was carried out in a sensory booth under controlled white incandescent lightening.

#### 2.5.8. Statistical Analysis

A completely randomized design was used for the entire experiment. All experiments were performed in triplicate. The data were evaluated using analysis of variance (ANOVA). Comparing means was performed using Duncan’s mean comparison. All statistical analyses were performed using the SPSS package (IBM SPSS Statistics 22.0, SPSS Inc., Chicago, IL, USA).

## 3. Results

### 3.1. PPO-IA and CCA of Various Dechlorophyllized Ethanolic Indigenous Leaf Extracts

The PPO-IAs of dechlorophyllized ethanolic noni, Jik, and soursop leaf extracts at varying levels are shown in Figure 1A. All the extracts had increasing PPO-IA with augmenting concentrations. Leaf extracts at 1% exhibited the highest inhibitory activity toward PPO (*p* < 0.05). At a 0.05% level, NLE and SLE showed no difference (*p* > 0.05) in PPO-IA, in which an approximately 18% inhibition was achieved. However, JLE showed the lowest inhibition (14.01%). Among the leaf extracts at a 1% concentration, SLE displayed the highest PPO-IA of 89.40% (*p* < 0.05), compared to 84.91% and 83.06% recorded for NLE and JLE, respectively. Olatunde et al. [14] reported that ethanolic guava leaf extracts (with and without chlorophyll removal) at varying concentrations (0.025–0.5) could inhibit PPO in Pacific white shrimp. PPO-IA was mainly caused by the high content of phenolic compounds in leaf extracts [14,20]. The phenolic compounds act as PPO inhibitors through interaction with the enzyme’s active site to prevent the oxidation of phenolic substrates or by the reduction of quinone, which could undergo further non-enzymatic polymerization to produce dark pigments [5]. It was also documented that catechin from green tea exhibited Pacific white shrimp PPO-IA [37], and catechin derivatives, especially epigallocatechin gallate (EGCG), were able to inhibit PPO of Pacific white shrimp via a mixed-type mechanism [38]. In addition, Liao et al. [39] found that salicylic acid could competitively inhibit PPO by binding to its active site through hydrogen bonding or hydrophobic interaction. This inhibition was also via the acidification effect of salicylic acid. Therefore, SLE had a higher potential for PPO inhibition than other extracts and was therefore employed to retard melanosis in refrigerated shrimp.

The CCA of all three leaf extracts is illustrated in Figure 1B. A significant increase in CCA was noticeable with the upsurge in extract concentrations (*p* < 0.05). At a 0.05% concentration, there was no difference in CCA among the three leaf extracts (*p* > 0.05). However, as the concentration of the extracts was augmented, SLE exhibited the highest increase in CCA compared to NLE and JLE. At the highest concentration (1%), SLE showed a higher CCA (65.30%) among all extracts. Coincidentally, the result of the CCA was in agreement with PPO-IA (Figure 1A). PPO is a copper-containing metalloprotein with a co-factor consisting of two copper atoms coordinated by three histidine residues each at its active site [8]. This site is essential for the enzyme catalytic activity in oxidizing phenolic compounds including tyrosine to quinones [5]. Metal-chelators such as EDTA, phosphates, kojic acid, and certain phenolic compounds like the flavonoids and phenolic acids present in plant extracts have been widely reported to form stable complexes with copper ions [9,40]. Sánchez-Vioque et al. [40] found that white thyme (*T. mastichina* L.) ethanolic extract obtained through Soxhlet extraction (SXEE) exhibited CCA with an IC_50_ of 112.7 μg/mL. The extract inhibited the enzyme with the ability to catalyze melanosis. In another study, catechin and ferulic acid showed higher copper chelation and could react with the intermediate browning reaction products, thereby preventing the formation of dopachrome in Pacific white shrimp [41]. SLE, exhibiting the highest PPO-IA and CCA, was selected as a promising candidate for further application in whole shrimp as an anti-melanosis agent to prevent melanosis. It could also display antimicrobial and antioxidant activities [30].

### 3.2. Identification of Phenolic Compounds in SLE

The phenolic compounds in SLE that exhibited the highest PPO-IA and CCA were identified using LC-MS profiling, as presented in Table 1. More than 23 phenolic compounds, predominantly phenolic acids and flavonoids, were determined in SLE based on molecular abundance, structure, and MS-spectra matching with the database [42]. Aempferol-3-O-rutinoside was the most abundant phenolic compound identified in SLE with a peak obtained at 11.15 min retention time. Other dominant phenolic compounds included catechin and neochlorogenic acid, respectively. Epicatechin, rutin, hyperin (quercetin-3-O-galactoside), kaemferol-7-O-neohesperidoside, isorhamnetin-3-O-neohespeidoside, p-coumaric acid, ferulic acid, caffeic acid, and others were also detected. The phenolic compounds are believed to be associated with potential biological activities, especially antioxidant, antimicrobial, and anti-melanosis activities. Sae-leaw et al. [32] documented the inhibitory effects of tea catechin and its four derivatives on PPO activity and melanosis in Pacific white shrimp. EGCG at 0.25–0.75% could control melanosis and inhibit PPO in a concentration-dependent manner. Other phenolic compounds with similar bioactivities were ferulic acid [17], neochlorogenic acid [43], rutin [44], kaemferol [45], etc. It was postulated that the individual or combined effects of the identified phenolic compounds could be responsible for PPO-IA and CCA of SLE.

### 3.3. Effect of SLE on Melanosis and Changes in the Quality of Pacific White Shrimp during the Refrigerated Storage

#### 3.3.1. Melanosis

The melanosis scores of the control and the samples treated with SLE at varying concentrations (SLE-0.25%, SLE-0.5%, SLE-1%) in comparison with the SMS-1.25% sample are depicted in Figure 2.

At day 0, the melanosis scores of all the treated samples and the control were approximately zero. The melanosis score rose with a longer storage time, particularly in the control, which had the highest score, compared to the treated samples (*p* < 0.05). Lower melanosis scores were recorded in samples treated with higher concentrations of the extract and those treated with SMS. The SMS-1.25% sample showed the lowest melanosis scores within the first 6 days of refrigerated storage (*p* < 0.05). As the storage time exceeded 9 days, the SLE-1% sample showed the lowest melanosis score (*p* < 0.05). The sharp upsurge in the melanosis score of the SMS-1.25% sample might be due to the instability of the sulfite compound as a result of the formation of sulfur dioxide, which reduced its potency in melanosis inhibition as the storage time progressed [46]. SMS can retard melanosis through the nucleophilic attack of sulfite at ο-quinone and by hydrogen addition, thereby reducing quinone in the reaction [47]. White shrimps were treated with catechin and its derivatives at various concentrations (0.25–0.75%) and stored at 4 °C for 10 days [32]. The SLE-1% sample exhibited the lowest melanosis score on day 12 of storage (*p* < 0.05). This was consistent with the highest PPO-IA and CCA of SLE, as presented in Figure 1A,B. The observed effect might be associated with high levels of phenolic compounds such as catechin, epicatechin, ferulic acids, rutin, etc., which were reported to inhibit melanosis in previous studies [41,48]. SLE could therefore inhibit melanosis in raw Pacific white shrimp during refrigerated storage at 4 °C. After 12 days of storage, the lowest melanosis was detected for the SLE-1% sample, whereas the control had the highest melanosis (Figure 3).

#### 3.3.2. Microbiological Changes

The microbiological quality of samples treated with SLE (0.25–1%) and SMS (1.25%) and the control, stored at 4 °C for 12 days, was monitored (Figure 4).

The initial TVC of the control and all treated shrimp samples ranged from 2.60 to 3.04 log CFU/g at day 0 (Figure 4A). At day 0, the control had slightly higher TVC (3.04 log CFU/g) than others, while similar TVC was observed among the treated samples (*p* > 0.05). During storage, a slight increase in TVC for all samples was attained within the first 3 days of storage. This might be due to the adaptation of mesophilic bacteria to the refrigeration temperature to a certain extent [49]. However, at a later stage, all samples had a continuous increase in TVC (*p* < 0.05). On day 9, the TVC of the control and 0.25% SLE-treated samples exceeded the microbiological limit (10^6^ CFU/g), rendering them unsuitable for human consumption. Nevertheless, the remaining treated samples had TVCs below the limit. At the end of the storage (day 12), only the SLE-1% sample had a TVC below the limit, while other samples exceeded the acceptable limit. TVC estimates mesophilic bacteria that grow at moderate temperatures, typically between 20 °C and 45 °C. They are commonly found in seafood and can cause spoilage and foodborne illness if not properly handled. Some common mesophilic bacteria found in seafood include *Aeromonas*, *Enterobacter*, and *Vibrio parahaemolyticus* [50]. The anti-bacterial properties present in soursop leaf extract might be caused by the presence of antimicrobial phenolic compounds such as hyperin, procyanidin A1, afzelin, quercitrin, chlorogenic acids, etc., which could inhibit bacterial growth by attacking the outer and plasma bacterial membranes [23]. Moreover, some phenolics like catechin and ferulic acid can chelate the metals that are vital for bacterial growth [41]. Therefore, mesophilic bacterial growth can be suppressed by the treatment of shrimp with 1% SLE, which demonstrated the lowest increase in TVC (*p* < 0.05).

PBC in Pacific white shrimp samples varied significantly (*p* < 0.05) in relation to the storage time, as illustrated in Figure 4B. On days 0 and 3, all samples exhibited a similar PBC of approximately 3.0 log CFU/g, and 3.5 log CFU/g, respectively and no differences in PBC were observed among the samples on the same day (*p* > 0.05). Subsequently, when the storage time reached 6 days and a longer time, different PBCs were noted among all the samples (*p* < 0.05). On day 12, the control sample had the highest PBC of 6.32 log CFU/g, followed by SLE-0.25% (6.07 log CFU/g), SMS-1.25% (5.98 log CFU/g), and SLE-0.5% (5.95 log CFU/g). The lowest PBC was observed in SLE-1% at 5.72 log CFU/g. The results suggested that psychrotrophic bacteria were the predominant microorganisms contributing to spoilage in seafood during cold storage. Psychrotrophic bacteria generally found in seafood include *Pseudomonas*, *Shewanella, and Photobacterium* [51]. The use of SLE at a concentration of 1% could be an effective means to prevent their growth and maintain refrigerated shrimp at a safe level for consumption for up to 12 days.

The initial *Pseudomonas* count was highest in the control (2.62 log CFU/g), while the sample treated with SMS revealed the lowest initial count at 2.41 log CFU/g (*p* < 0.05) (Figure 4C). Nonetheless, a similar *Pseudomonas* count was noticeable among the SLE-treated samples (*p* > 0.05). The observed result exhibited a similar trend with TVC and PBC. This could be attributed to the inability of the bacteria to adapt themselves to the new environmental conditions associated with storage. Moreover, the *Pseudomonas* count in all samples increased (*p* < 0.05) as the storage time exceeded 3 days. Notably, SLE-treated samples exhibited a lower increase in *Pseudomonas* counts in a dose-dependent manner. At the end of the storage period, the samples treated with SLE (0.5–1%) and SMS (1.25%) showed low *Pseudomonas* counts, whereas the control and SLE-0.25% samples demonstrated the highest counts. *Pseudomonas* spp. is the predominant specific spoilage organism in many seafoods. High-throughput sequencing analysis showed that *Pseudomonas* accounted for approximately 83% of the flora in grass carp fillets treated with lysozyme and phytic acid and stored at 4 °C for 8 days [52]. In another study, it was revealed that *Pseudomonas,* classified as psychrotrophic bacteria, was capable of thriving under refrigeration conditions by utilizing glucose and amino acids obtained from the seafood, which consequently led to spoilage [53]. Overall, SLE at concentrations of 0.5% to 1% was found to be more effective than SMS at 1.25% for inhibiting the growth of *Pseudomonas* spp. to a safe level during storage at 4 °C.

The Enterobacteriaceae count gives an estimate of several gram-negative bacteria species associated with seafood-borne illnesses [54,55]. Several species such as *E. coli*, *Salmonella*, *Shigella*, *Klebsiella*, and *Vibrio parahaemolyticus* belong to this group [56]. It is imperative to keep this group of bacteria at a low level in raw seafood to ensure food safety and prevent foodborne illness. At day 0, the Enterobacteriaceae count of the treated samples and the control ranged from 2.19 to 2.33 log CFU/g (Figure 4D). The bacterial counts of all the treatments and the control were lower, except for the SLE-0.25% sample, which showed a slightly higher value than others. This could be due to minor variations in the initial bacterial load among the samples. At day 3, an increase in the Enterobacteriaceae count was found for all samples (*p* < 0.05). By the end of the storage (day 12), the lowest increase in Enterobacteriaceae count was observed in the SLE-1% sample with a value of 4.31 log CFU/g. In contrast, the control had the highest increase in Enterobacteriaceae count, reaching 4.94 log CFU/g, followed by the SLE-0.25% sample. The SMS-1.25% and SLE-0.5% samples had no differences in Enterobacteriaceae count (*p* > 0.05). Hence, it can be inferred that the treatment of shrimp with SLE at 1% could efficiently lower the growth of Enterobacteriaceae during refrigerated storage.

The H_2_S-producing bacteria counts at day 0 of storage ranged from 2.36 to 2.59 log CFU/g (Figure 4E). Similar to the TVC, PBC, and *Pseudomonas* count, there was a continuous growth of H_2_S-producing bacteria until day 12 (*p* < 0.05). The bacterial growth was retarded, and the inhibition depended on the concentration of SLE. Accelerated growth was observed in the control. H_2_S-producing bacteria constitute a diverse group of microorganisms that generate hydrogen sulfide as a metabolic byproduct, thereby contributing to off-odor in seafood [53]. This is often related to the formation of putrescine and indole, which are the major indicators of shrimp spoilage. In addition, the off-odor is generated by the released gases such as H_2_S and NH_3_ [53]. Several plant extracts containing various kinds of phenolic compounds including cashew [11], guava [56], Jik [30], and Chamuang [46] have been reported to exhibit inhibitory effects on the growth of H_2_S-producing bacteria in shrimp during cold storage. Thus, SLE at 1% could be considered a promising candidate for effectively inhibiting the growth of H_2_S-producing bacteria.

#### 3.3.3. Chemical Changes

##### pH

The pH values of the samples at day 0 ranged from 6.51 to 6.60 (Figure 5A). This value was within the acceptance limit (7.6) for shrimp [46,57]. As the storage time progressed, the pH values began to rise, especially in the control. This result indicated the production of volatile base substances as a result of autolysis or bacterial spoilage associated with protein decomposition [4]. Higher pH was coincidental with the increase in *Pseudomonas* count and H_2_S-producing bacteria count, as shown in Figure 4C,E, respectively. At day 12, the lowest pH was observed in SLE-1% samples, whereas the control had the highest pH (*p* < 0.05). Samples treated with SLE at lower concentrations and SMS-1.25% displayed a moderately lower pH increase. These findings were consistent with those documented by Olatunde et al. [14], who applied an ethanolic guava leaf extract dechlorophyllized by sedimentation to extend the shelf-life and inhibit microbial growth in Pacific white shrimp. In general, SLE at 1% showed a potential natural preservative function that could retard microbial growth and slow down decomposition of protein or non-protein nitrogenous compounds, thereby reducing the rate of pH increase due to the generation of volatile base compounds.

##### TVB-N Content

The initial TVB-N content of the shrimp samples ranged from 6.35 to 7.75 mg N/100g of shrimp meat (Figure 5B). At day 0, there was no difference in TVB-N content between the control and those of 0.25% and 0.5% SLE-treated shrimps (*p* > 0.05). Likewise, no variation between the TVB-N content of the SLE-1% and SMS-1.25% samples was noted (*p* > 0.05). As the storage continued, there was a constant increase in TVB-N among all the samples, but a lower increase was noticed in SLE (0.5% and 1%) and SMS-1.25% shrimps (*p* < 0.05). The control and SLE-0.25% shrimps had the highest increase in TVB-N content. At day 12, the SLE-1% sample possessed the lowest TVB-N content (19.1 mg N/100 g shrimp meat). The observed results might be attributed to the high concentration of SLE, which could reduce bacterial growth associated with the deamination of non-protein nitrogenous compounds [58]. An increase in TVB-N values was also correlated with higher bacterial loads (Figure 4) and pH (Figure 5A) of the shrimp meat, as indicated by the production of basic nitrogenous compounds [59]. TVB-N is an essential indicator of bacterial spoilage and is a product of protein breakdown. An upsurge in TVB-N level is related to bacterial spoilage and endogenous enzyme activity [50]. As reported by Shiekh et al. [46], the freshly caught shrimp typically had acceptable TVB-N below 12 mg N/100g. TVB-N levels ranging from 12 to 20 mg N/100g are considered suitable for consumption. However, TVB-N contents between 20 and 25 mg N/100g are considered the tolerance limit, indicating a potential decline in quality. TVB-N content greater than 25 mg N/100g is considered “inedible and decomposed” [60]. Therefore, it can be deduced that the SLE treatment of shrimp at 1% prior to storage at 4 ˚C could inhibit spoilage bacterial growth, responsible for the formation of undesirable volatile basic nitrogen compounds in shrimp during storage.

##### TBARS Value

At day 0, the TBARS values of all samples ranged from 0.6 to 1.01 mg MDA equivalent/kg of shrimp meat. As storage advanced, there was an upsurge in TBARS values with increasing storage time (*p* < 0.05) (Figure 5C). The control samples possessed the highest increase in TBARS, compared to SMS and SLE-treated samples at all storage times. Nevertheless, the samples treated with SLE had a lower increase in TBARS value and the rate of increase was lower with increasing SLE concentrations. As the storage ended, all samples showed higher TBARS values (2.42–4.21 mg MDA equivalent/kg shrimp), and the control had the highest value (*p* < 0.05). SLE-1% sample had the lowest TBARS (*p* < 0.05). The tissue membranes of crustaceans have high levels of PUFA, which can undergo lipid oxidation if the tissue is damaged during handling or processing [5]. The TBARS value is a widely used measure of secondary lipid oxidation associated with oxidative stress and lipid peroxidation in biological substances. Since SLE is rich in phenolic compounds that are linked with antioxidant activities, it is believed to prevent lipid oxidation in shrimp by retarding the formation of unstable hydroperoxides that might be generated during storage, which could be decomposed to malonaldehyde. Moreover, it has been reported that soursop leaf extracts scavenged DPPH and ABTS radicals, had reducing power and chelated metal ions [30,61]. Overall, the treatment of shrimp with 1% SLE could profoundly retard the rate of lipid peroxidation in shrimp during refrigerated storage.

#### 3.3.4. Sensorial Changes

The likeness scores of mildly cooked shrimp samples treated with SLE (0.25–1%) and SMS-1.25% and the control at day 0 and the SLE-1% sample kept for 12 days of refrigerated storage, which had a TVC lower than the microbiological limit (10^6^ CFU/g), are presented in Table 2. At day 0, similar likeness scores were obtained in the control and all SLE-treated samples, irrespective of SLE concentrations (*p* > 0.05). The mean scores ranged from 7.50 to 8.40, which was interpreted as “Liked very much” on the 9-point hedonic scale. It was evident that the SMS-1.25% and SLE (0.25–1%) solutions did not impact negatively on the sensory attributes of the pre-cooked shrimp samples. At day 12, the scores of all sensory indices of the SLE-1% cooked shrimp sample were decreased (*p* < 0.05). This was in line with the high TVB-N and TBARS values recorded in SLE-1% shrimp sample (Figure 5). The overall acceptability score of SLE-1% shrimp kept for 12 days was 6.8, indicating “liked moderately” samples. Also, the lower likeness color score could be caused by melanosis taking place to some degree in the SLE-1% shrimp (Figure 2). Therefore, 1% SLE treatment on shrimp could preserve its sensory qualities during refrigerated storage.

## 4. Conclusions

Soursop leaf ethanolic extract after dechlorophyllization served as a promising alternative to synthetic SMS for inhibiting melanosis and preserving the quality of Pacific white shrimp during refrigerated storage. This was more likely because of phenolic compounds in the extract, in which aempferol-3-O-rutinoside, catechin, and neochlorogenic acid were abundant compounds, among other phenolic compounds. The inhibitory activity of SLE-1% against PPO, the high CCA, the retardation of bacterial growth, the deceleration of chemical deterioration, and the better sensory evaluation scores collectively resulted in an extended shelf-life of 12 days as compared to 6 days for the control sample. Further research is warranted to explore the potential application of SLE in seafood products and other food products.

## Figures and Tables

**Figure 1 foods-12-03649-f001:**
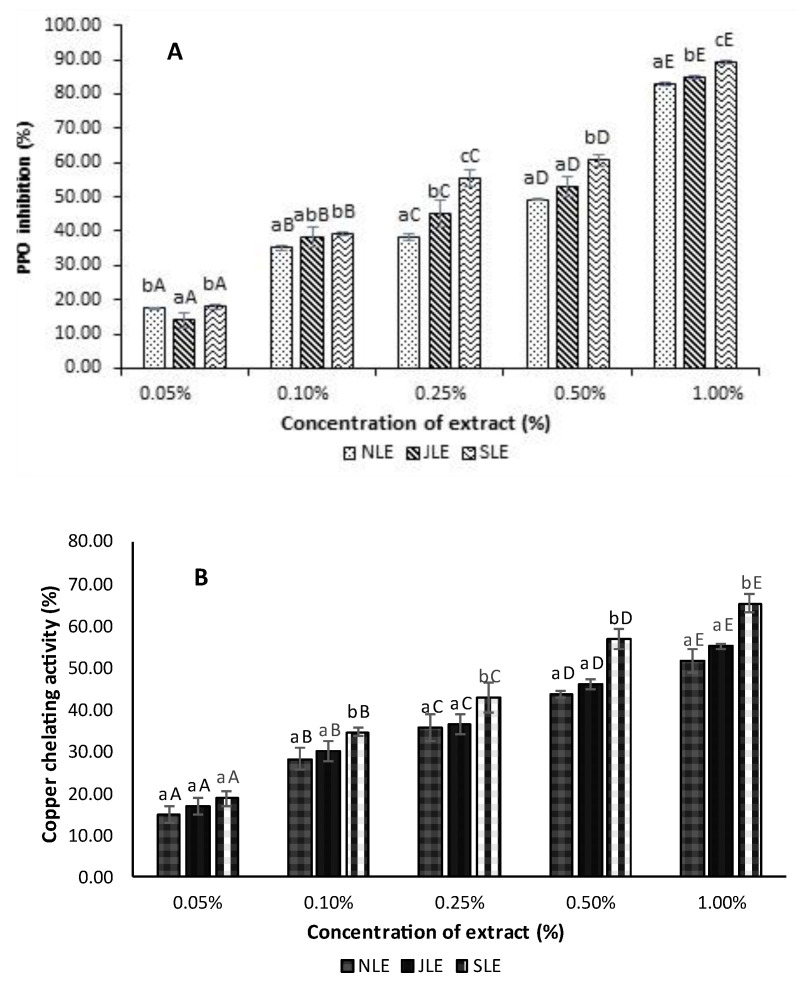
The PPO inhibitory activity (**A**) and copper-chelating activity (**B**) of dechlorophyllized ethanolic extracts from three Thai indigenous leaves. Bars represent the standard deviation (*n* = 3). Different lowercase letters on the bars within the same concentration indicate significant differences (*p* < 0.05). Different uppercase letters on the bars within the same leaf extract indicate significant differences (*p* < 0.05).

**Figure 2 foods-12-03649-f002:**
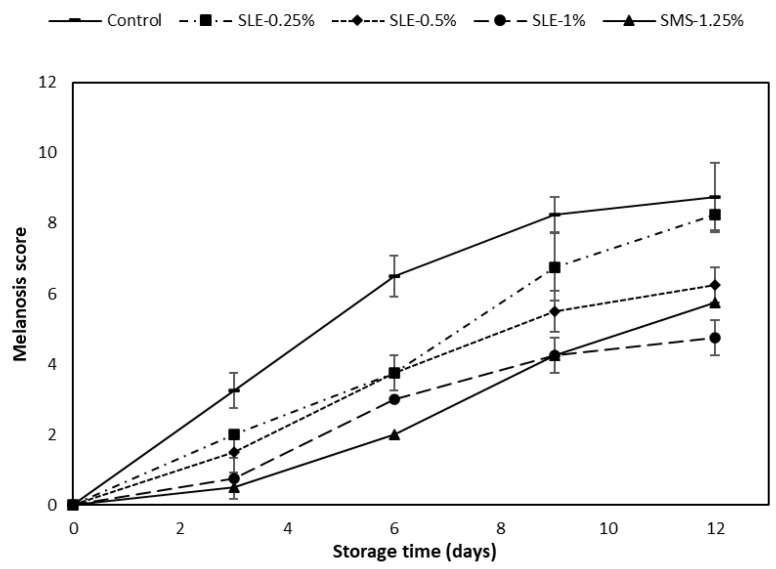
Melanosis scores of Pacific white shrimps without and with treatment using dechlorophyllized ethanolic soursop leaf extract at different concentrations during 12 days of refrigerated storage. Bars represent the standard deviation (*n* = 10). SLE-0.25%, SLE-0.5%, and SLE-1%, represent Pacific white shrimp treated with SLE at 0.25, 0.5, and 1%, respectively. Control and SMS-1.25% represent Pacific white shrimp treated with distilled water and 1.25% sodium metabisulfite, respectively.

**Figure 3 foods-12-03649-f003:**
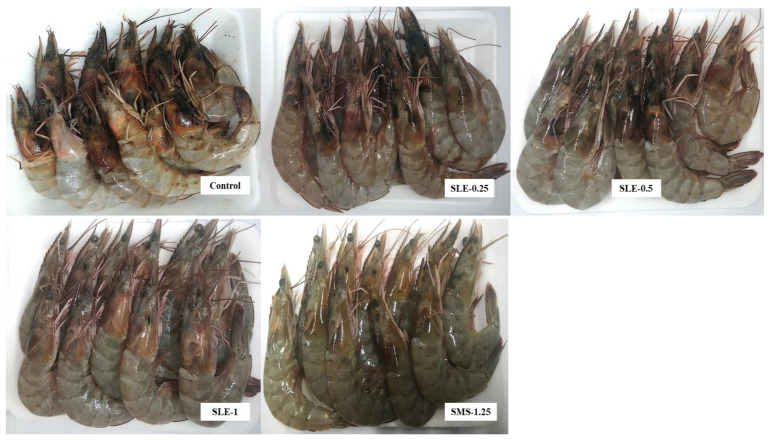
Photographs of Pacific white shrimp without and with treatment using dechlorophyllized ethanolic soursop leaf extract at different concentrations after 12 days of refrigerated storage.

**Figure 4 foods-12-03649-f004:**
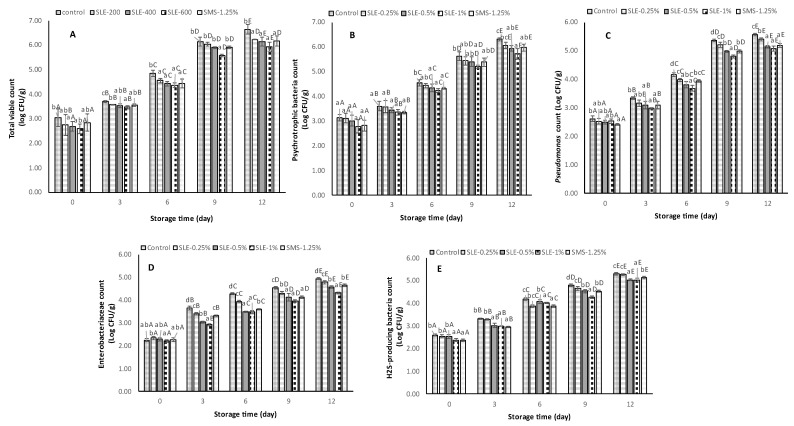
Total viable count (**A**), psychrotrophic bacterial count (**B**), *Pseudomonas* count (**C**), *Enterobacteriaceae* count (**D**), and H_2_S-producing bacterial count (**E**) of Pacific white shrimp treated with dechlorophyllized ethanolic soursop leaf extract (SLE) at different concentrations over 12 days of refrigerated storage. Bars represent the standard deviation (*n* = 3). Different lowercase letters on the bars within the same storage time indicate significant differences (*p* < 0.05). Different uppercase letters on the bars within the same treatment indicate significant differences (*p* < 0.05). SLE-0.25%, SLE-0.5%, and SLE-1%, represent Pacific white shrimp treated with SLE at 0.25, 0.5, and 1%, respectively. Control and SMS-1.25% represent Pacific white shrimp treated with distilled water and 1.25% sodium metabisulfite, respectively.

**Figure 5 foods-12-03649-f005:**
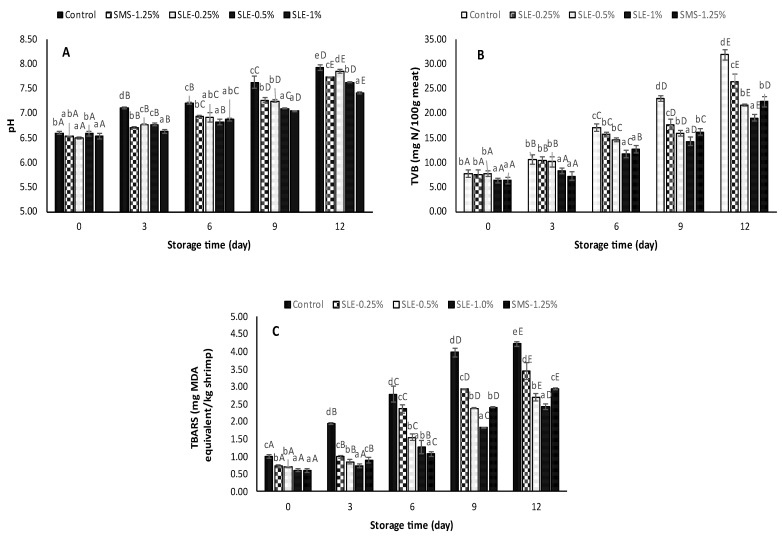
The pH (**A**), total volatile base (TVB) compounds (**B**), and thiobarbituric acid reactive substances (TBARS) (**C**) of Pacific white shrimp treated with dechlorophyllized ethanolic soursop leaf extract (SLE) at different concentrations during 12 days of refrigerated storage. Bars represent the standard deviation (*n* = 3). Different lowercase letters on the bars within the same storage time indicate significant differences (*p* < 0.05). Different uppercase letters on the bars within the same treatment indicate significant differences (*p* < 0.05). SLE-0.25%, SLE-0.5%, and SLE-1% represent Pacific white shrimp treated with SLE at 0.25, 0.5, and 1%, respectively. Control and SMS-1.25% represent Pacific white shrimp treated with distilled water and 1.25% sodium metabisulfite, respectively.

**Table 1 foods-12-03649-t001:** LC-MS profile of phenolic compounds identified in dechlorophyllized ethanolic soursop leaf extract.

Compounds	Molecular Weight	Molecular Formula	Score	Abundance (×10^6^)
Aempferol-3-O-rutinoside	594.16	C₂₇H₃₀O₁₆	94.9	60.65
Catechin	290.08	C₁₅H₁₄O₆	97.6	25.31
Neochlorogenic acid	354.1	C₁₆H₁₈O₉	98.7	18.39
Rutin (quercetin 3-rutinoside)	610.14	C₂₇H₃₀O₁₆	94.6	14.86
Hyperin (quercetin-3-O-galactoside)	464.1	C₂₁H₂₀O₁₂	99.6	12.56
Kaemferol-7-O-neohesperidoside	594.16	C₂₉H₃₆O₁₇	83.1	9.99
Isorhamnetin-3-O- neohespeidoside	578.16	C₃₃H₄₈O₁₇	96.8	9.95
Procyanidin C1	866.21	C₃₀H₂₄O₁₂	96	8.28
Quercitrin	448.1	C₂₁H₂₀O₁₁	71.4	5.53
Afzelin	432.11	C₂₁H₂₀O₁₁	99.6	4.35
Cryptochlorogenic acid	354.1	C₁₇H₂₀O₉	98.2	4.2
Epicatechin	290.08	C₁₅H₁₄O₆	96.4	3.39
Procyanidin B2	578.14	C₃₀H₂₄O₁₂	97	3.25
Procyanidin A1	576.13	C₃₀H₂₄O₁₁	76.3	1.58
Calceorioside B	478.21	C_23_H_26_O_11_	94.6	1.1
p-Coumaric acid	164.05	C₉H₈O₃	98.3	0.75
Phloridzin (dihydrochalchone)	436.14	C₂₁H₂₄O₁₀	78.9	0.55
Salidroside	300.12	C₁₄H₂₀O₇	90.9	0.54
Caffeic acid	180.04	C₉H₈O₄	92.2	0.53
Chlorogenic acid	354.1	C₁₆H₁₈O₉	98.7	0.48
Protocatechin aldehyde	138.03	C₉H₈O₂	98.9	0.229
Isoferulic acid	194.06	C₁₀H₁₀O₄	93.3	0.22
Ferulic acid	194.06	C₁₀H₁₀O₄	94.5	0.19

**Table 2 foods-12-03649-t002:** Likeness scores of pre-cooked Pacific white shrimps treated with dechlorophyllized ethanolic soursop leaf extracts at different concentrations at days 0 and 12 of refrigerated storage.

Storage Time (Days)	Sample	Appearance	Color	Texture	Taste	Flavor	Odor	Overall Likeness
0	Control	8.00 ± 0.71 a	8.30 ± 0.71 a	7.50 ± 0.93 a	7.75 ± 0.38 a	8.11 ± 0.60 a	7.56 ± 0.53 a	7.89 ± 0.60 a
	SLE-0.25	8.20 ± 0.96 a	8.20 ± 0.46 a	8.00 ± 0.71 a	7.60 ± 0.58 a	8.00 ± 0.71 a	8.20 ± 0.96 a	8.00 ± 0.71 a
	SLE-0.5	7.60 ± 0.89 a	8.20 ± 0.84 a	7.80 ± 0.45 a	8.00 ± 0.71 a	8.20 ± 0.84 a	7.80 ± 0.84 a	7.80 ± 0.84 a
	SLE-1.0	8.10 ± 0.74 a	8.20 ± 0.45 a	7.60 ± 0.55 a	7.60 ± 0.55 a	8.00 ± 0.71 a	8.20 ± 0.45 a	7.60 ± 0.55 a
	SMS-1.25	8.40 ± 0.55 a	8.40 ± 0.55 a	7.80 ± 0.84 a	7.80 ± 0.84 a	8.20 ± 0.45 a	8.00 ± 0.71 a	7.60 ± 0.55 a
12	SLE-1.0	7.2 ± 0.84 b	7.00 ± 0.55 b	7.10 ± 0.84 b	6.20 ± 0.84 b	7.00 ± 0.71 b	6.40 ± 0.89 b	6.80 ± 0.84 b

* Values represent the mean and standard deviation (*n* = 50). Different lowercase letters in the same column indicate significant differences (*p* < 0.05). SLE-0.25%, SLE-0.5%, and SLE-1% represent shrimps treated with 0.25, 0.5, and 1% SLE. Control and SMS-1.25% represent shrimp treated with distilled water and 1.25% sodium metabisulphite, respectively.

## Data Availability

The data used to support the findings of this study can be made available by the corresponding author upon request.

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
