# Peer review of "Impact of Ethanolic Thai Indigenous Leaf Extracts on Melanosis Prevention and Shelf-Life Extension of Refrigerated Pacific White Shrimp"

_foods, 2023, doi:10.3390/foods12193649_

Round 1
Reviewer 1 Report
This manuscript described the effects of extracts from soursop leaves on shelf-life extension of refrigerated Pacific white shrimp. This manuscript was written well. However, some details need to be added. Here are the reviewer’s detailed comments. Please consider the reviewer’s opinion in revising this manuscript to make it more comprehensive.
General and specific comments:
1. Introduction: Please provide additional information about Noni (Morinda citrifolia) and Jik (Barringtonia acutangula).
2. Line 173: Why was the shrimp treated with 1.25% SMS? Was the SMS solution concentration too high?
3. Fig.3: What is the reason for the shrimp's body being white in the control group but black in the other groups. What is the reason for the shrimp's body being blackest in the SMS group?
4. Please provide more data on sensory evaluation, especially for all groups on the 10th and 12th day.
Author Response
Response to reviewer
****Thank you for the valuable comments and suggestions. All the queries have been responded and the corrections have been made where applicable as highlighted in yellow.
General and specific comments:
- Introduction: Please provide additional information about Noni (Morinda citrifolia) and Jik (Barringtonia acutangula).
****Details regarding Noni and Jik have been provided in the introduction. Please see lines 81 to 95
- Line 173: Why was the shrimp treated with 1.25% SMS? Was the SMS solution concentration too high?
****SMS at 1.25% is commonly used by shrimp farmers due to its effectiveness against microorganisms and its ability to reduce melanosis. The concentration was chosen to reflect what has been used in the market and then compared with the plant extracts to achieve the same goal. However, SMS is not safe for consumption, especially for people who are allergic to sulfites, and asthmatics.
FDA's allowable limit of sulfite in foods was set at 500 ppm, based on the SO2 residue and allowable daily intake (ADI) of 0.7 mg/kg bw/day. SMS undergoes several reactions, whose stability as a food preservative could be reduced during prolonged storage due to various factors including conversion into SO2 gas, change in pH, oxidation to sulfate and sulfite ions, etc. Hence the final concentration of SO2 equivalent is generally less than 1%.
- Fig.3: What is the reason for the shrimp's body being white in the control group but black in the other groups? What is the reason for the shrimp's body being blackest in the SMS group?
****PPO, responsible for blackening, is primarily located in the cephalothorax, followed by the abdominal exoskeleton. It usually spreads to the telson and uropods when melanosis is further developed.
In the control group, the highest blackspots were observed in the cephalothorax and telson regions. However, melanosis appeared very low in the cephalothorax region and the whole body of SMS-treated group, compared to the control at the early stage of storage. It later increased rapidly and spread as storage progressed. This could be due to the instability of SO2, thus reducing SMS effectiveness. The discussion had been already made in line 333-336.
- Please provide more data on sensory evaluation, especially for all groups on the 10th and 12th day.
****Sensory evaluation was carried out only at the beginning (Day 0) and at the end (Day 12) of the storage period. Only SLE-1 % treated shrimps were assessed on day 12 because other samples had exceeded the safe microbial limit of 6 Log CFU/g. The reason has been given for selecting SLE-1 % treated shrimps stored for 12 days for sensory evaluation. Please see line 516.

Reviewer 2 Report
The document evaluates the antioxidant activities of three plants used in Thailand for food preparation or in traditional medicine. Later, one is selected to further study its use to increase the shelf life of white shrimp.
It is convenient to include the yield of the extracts to determine the concentration needed of the plant to achieve the results shown by the extracts.
In figure 4 it would be really useful if the maximum allowed microbial concentration is permitted or indicates that the product is still microbiologically safe.
Do panelists in the sensory analysis were trained?
The document needs a revision by a language editor.
Need moderate reivision
Author Response
Response to reviewer
****Thank you for the valuable comments and suggestions. All the suggestions have been responded and the corrections have been made where applicable as highlighted in green.
- It is convenient to include the yield of the extracts to determine the concentration needed by the plant to achieve the results shown by the extracts.
****Details including the yield, total phenolic content, total flavonoid content and MIC/MBC values were given in our previous studies. Antioxidant and antimicrobial properties of these leaf extracts were also studied (Ahmad et al, 2023). The aforementioned data, especially yield, were considered in formulating the concentrations of the extracts used in the present study.
- In figure 4 it would be really useful if the maximum allowed microbial concentration is permitted or indicates that the product is still microbiologically safe.
****The maximum allowable limit (6 Log CFU/g) for total bacterial count in raw seafood was stated. Please see line 361.
- Do panelists in the sensory analysis were trained?
****Since the likeness for the selected quality attributes including appearance, color, texture, taste , flavor, odor and overall acceptability was tested. Fifty untrained panelists were involved in the sensory evaluation. The information has been given in line 232.
- The document needs a revision by a language editor.
****Well noted. English has been checked throughout the manuscript using the ‘Grammarly’ software.

Round 2
Reviewer 1 Report
The manuscript has been improved appropriately.